# Fever of Unknown Origin and Penetrating Aortic Ulcer Successfully Treated with Thoracic Endovascular Aortic Repair—A Case Report

**DOI:** 10.3390/diagnostics15233077

**Published:** 2025-12-03

**Authors:** Tomislav Jakljević, Franka Kunovac, Tatjana Zekić, Vjekoslav Tomulić

**Affiliations:** 1Clinic for Cardiovascular Diseases, Clinical Hospital Center Rijeka, Tome Strižića Street 3, 51 000 Rijeka, Croatia; vjekoslav.tomulic@uniri.hr; 2Department of Rheumatology and Clinical Immunology, Internal Medicine Clinic of Clinical Hospital Center Rijeka, Krešimirova Street 42, 51 000 Rijeka, Croatia; franka.kunovac1@gmail.com (F.K.); zekic.tatjana79@gmail.com (T.Z.)

**Keywords:** atherosclerotic ulcer, penetrating, aortitis, endovascular aneurysm repair, fever of unknown origin

## Abstract

**Background and Clinical Significance**: Fever of undetermined origin (FUO is a diagnostic challenge. It is essential to exclude infections, paraneoplastic syndromes, and large-vessel vasculitis (LVV). **Case presentation**: We describe a 59-year-old female with FUO and no apparent signs of infection. Laboratory results were consistent with inflammation (ESR 83, CRP 203 (ref. value (RV) < 5 mg/dl), ferritin 311 (RV < 120 µg/L), microcytic anemia, thrombocytosis. With administration of both ceftriaxone and levofloxacin, a decrease in CRP was monitored (51 mg/L). HRCT of the chest, abdomen, and pelvis revealed a saccular aneurysm of the descending thoracic aorta and an ectatic right common iliac artery. Due to suspicion of LVV, CT angiography was performed to exclude inflammatory changes in the blood vessels. Diffuse atherosclerosis with a 30 mm penetrating thoracic aortic ulcer (PAU) was found. PET-CT and leukocyte scintigraphy were used to rule out vasculitis and infection. The patient was successfully treated with Thoracic Endovascular Aortic Repair (TEVAR). **Conclusions:** As sophisticated imaging techniques become more widely used, more PAUs are being detected as incidental abnormalities in individuals without acute aortic syndrome. With adequate management, many people with PAU can live a stable and healthy life without experiencing significant consequences.

## 1. Introduction

The most common large-vessel vasculitis (LVV) after age 50 is giant cell arteritis (GCA). The pooled incidence of GCA is 10 cases per 100,000 adults over 50. The overall pooled mortality rate from GCA is 20 cases per 1000 persons [1]. Aortic aneurysms can develop in 6 to 20% of GCA patients, particularly in the ascending thoracic aorta and resulting in a substantial increase in mortality (standardized mortality ratio 2.83) [2]. Computed tomography angiography (CT angiography) is considered the first-line diagnostic technique because it is readily available and enables differential diagnosis from other causes of chest discomfort [3]. The estimated incidence of Penetrating aortic ulcer (PAU) ranges from 0.3 to 2.1 per 100,000 person-years, but precise, accurate data are unavailable. Penetrating aortic ulcers occur in around 7.5% of acute aortic syndromes and are more prevalent in older males with risk factors such as hypertension, smoking, and atherosclerosis [4,5]. Aortitis in the setting of vasculitis is usually accompanied by general symptoms such as fever and malaise, and may trigger aneurysm, aortic dissection, or rupture [6]. PAUs are a specific type of ulceration that occurs when atherosclerotic plaques erode the aortic wall. In the work-up of FUO, it is necessary to exclude infectious agents, autoimmune diseases, and paraneoplastic syndrome.

The literature shows cohorts of patients with PAU who were asymptomatic or presented with acute thoracic syndrome. Case reports on SCOPUS (search term “penetrating aortic ulcer and fever,” date 30 September 2025) present 34 retrievals, mostly infection as the leading cause of aortitis.

We present a case of PAU that presented with fever and as an LVV mimicker, which was successfully treated with Thoracic Endovascular Aortic Repair (TEVAR).

## 2. Case Description

### 2.1. History

A 59-year-old female was treated for fever of unknown origin FUO by an infectious disease specialist for 15 days without clear evidence of an infection. She was then transferred to the Department of Immunology. Her previous medical history included arterial hypertension and femoro-popliteal bypass of the left leg due to occlusion of the left superficial femoral artery. She is a smoker and obese, with a BMI of 41 kg/m^2^. She has been smoking for about 40 pack/years. Her previous treatment included anticoagulant (warfarin) and a combination of an ACE inhibitor (perindopril 4 mg) with a calcium channel blocker (amlodipine 5 mg). Upon admission, the following lab results were noteworthy: ESR 83, CRP 138.9 (highest pre-hospital value CRP 203 (ref. value < 5 mg/L), ferritin 311 (ref.value < 120 µg/L), microcytic anemia (Hgb 100 g/L), thrombocytosis (Trc 467 [1 × 10^9^ L]), normal leukocytes with mild neutrophilia (Leu 8.6 [1 × 10^9^]/L, Neu 6.13 [1 × 10^9^]/), normal urin analysis. Reuma factor (RF) (negative < 14 U/L), antinuclear antibody (ANA) (negative < 1:100), and antinuclear cytoplasmic antibody (ANCA) findings were negative. With intravenous (IV) antibiotics, ceftriaxone 2 g and levofloxacin 500 mg, a decline in CRP was observed (51 mg/L). After this rapid decline, the subsequent decrease in CRP was very slow; specifically, over 6 weeks, the values decreased to 14 mg/L. Blood and urine cultures, calprotectin, and serology for hepatitis, Epstein–Barr, and Cytomegalovirus (HBV, HCV, EBV, and CMV) were all normal.

### 2.2. Imaging

The patient complained of thoracolumbar back pain both at rest and with effort. An X-ray of the thoracic and lumbar spines showed reduced thoracic kyphosis and lumbar lordosis, with a mild S scoliotic curve. The vertebral bodies were normal in height, with spondylotic changes. The dorsal intercorporal line was slightly stepped with a grade I ventrolysis of the L4 vertebral body. Calcifications of vascular structures were seen. The chest, abdomen, and pelvis CT revealed a saccular aneurysm of the descending thoracic aorta, an ectatic right common iliac artery, reactive lymph nodes, and an adenoma of the right adrenal gland. Due to suspicion of large vessel vasculitis, CT angiography of the brain, carotids, and aorta was performed to exclude inflammatory changes in the blood vessels. Diffuse atherosclerosis with a 3 cm penetrating thoracic aortic ulcer (PAU) was demonstrated (length 30 mm, lumen width 13 mm), along with dilatation of the descending thoracic aorta proximal and distal to the PAU (Figure 1). The lumen width of both carotid arteries (ACC, ACI, and ACE) was normal. No aneurysmal formations or AV malformations were observed. Also, there was no visible occlusion of the large cerebral arteries.

Leukocyte scintigraphy after antibiotic therapy did not reveal pathological accumulations of autologous Tc99m-HMPAO-labeled leukocytes. Furthermore, PET CT showed no signs of inflammation, vasculitis, or paraneoplastic syndrome, but did not describe PAU either. Both examinations excluded infection and vasculitis, and TEVAR could be performed.

### 2.3. Management

After being transferred to the Department for Interventional Cardiology, the patient underwent successful implantation of a thoracic stent graft (Figure 2). In the later course of TEVAR, a drop in the blood count was monitored with mild subcutaneous bleeding in the left femoral region. CT aortography ruled out bleeding as well as free fluid or a possible hematoma in the abdomen and pelvis. A thin, elongated pseudoaneurysm was found in the left inguinal region. Posthemorrhagic anemia was partially corrected with a single dose of erythrocyte concentrate. Local bleeding in the left inguinal region was stopped with slightly stronger manual compression. A feverish episode and increased inflammatory markers prompted the initiation of empirical antibiotic therapy. In the subsequent course, the inflammatory markers declined. By the time of release, the patient had hematomas at the puncture site in resorption, was afebrile, clinically cardiac-compensated, asymptomatic, normotensive, and had a regular course of mobilization.

## 3. Discussion

Our case shows PAU with fever of unknown origin as a mimicker of LVV, successfully treated with antibiotics and thoracic endovascular aortic repair. There are several mimickers of LVV, such as chronic infection (tuberculosis, syphilis, *Coxiella burnetti*), IgG4-related disease, Erdheim Chester disease, RA, SpA, SLE, sarcoidosis, and other primary vasculitides (Behçet disease, relapsing polychondritis, or VEXAS syndrome), atherosclerosis, and malignancy [7]. Aortic dissection, intramural hematoma, and penetrating atherosclerotic ulcer are among the aortic diseases that can be quickly ruled out by CT, which is commonly available in most medical settings and typically involves iodinated contrast (CTA) [8]. In a study of 171 patients with GCA aortitis, 55 patients (32%) reported aortitis symptoms at diagnosis, including dorsal/lumbar/abdominal discomfort and aortic insufficiency. Aortic complications developed after a median of 32 months, with a median follow-up of 38 months. Five dissections and 19 new aortic aneurysms or aneurysm complications were reported [9]. In our case report, the patient also complained of back pain and, given the fever and the gradual decline in CRP, vasculitis was suspected.

### 3.1. Differences Between LVV-GCA and PAU

Currently, no specific diagnostic test is available to confirm LVV-GCA, as biopsy of these arteries is rarely performed in routine clinical practice, except during surgical intervention. LVV-GCA is typically suspected based on indirect evidence of large-vessel vasculitis on vascular imaging, such as CT angiography, 18-FDG PET, or MR angiography. The aorta is the main site of inflammation, followed by the carotid and subclavian arteries. Diagnosis of LVV-GCA is highly probable when clinical features of polymyalgia rheumatica (PMR) are present [10]. In contrast, the main criterion for PAU diagnosis is a localized collection of contrast material that extends beyond the lumen. According to several studies, PAU is most commonly seen in the descending aorta (61.2%), followed by the abdominal (29.7%) and arch (6.8%) regions. There are many PAU cohorts of 18–100 people published without acute thoracic syndrome as an accidental finding [4]. Asymptomatic PAU can be diagnosed on a PET-CT scan [11]. In our case, PAU was not identified on PET-CT despite measuring 30 mm. Still, the fortunate circumstance was that the patient had a fever, which led to suspicion of LVV and appropriate imaging.

PAU may manifest with multiple ulcers of varying diameters and depths; however, treatment is indicated when it extends more than 20 mm deep [12]. According to the literature, the most common pathological finding in the series of aortic specimens was medial degeneration, which was equally common in individuals over and under 65 [13]. A case with simultaneous LVV-GCA and PAU is described. An 87-year-old patient had intimal protrusion of the thoracic aorta, indicating an ulcer-like projection (ULP). Positron emission tomography-CT revealed a diffuse buildup of fluorodeoxyglucose in the aorta wall, including the ULP [14].

### 3.2. Treatment

Small, isolated, and asymptomatic PAUs can be treated conservatively with regular surveillance, which is associated with a lower risk of aortic-related death. When compared to medical care alone, both open surgery and TEVAR have been linked to decreased mortality rates, especially for high-risk or symptomatic lesions [15]. Despite the need for arch stent grafting, TEVAR is a safe elective surgery for treating aortic PAU. Early intervention at smaller aortic diameters (<55 mm) may benefit selected patients’ late survival [16]. Surgical intervention is needed in case of rupture or impending rupture. Signs of rupture include hemodynamic instability, shock, or hemorrhagic pleural effusion. Impending rupture: symptoms include severe recurrent pain accompanied by a periaortic hematoma, or evidence of ongoing bleeding into the aorta or surrounding tissues [16,17].

To identify potential problems in asymptomatic patients who underwent TEVAR, contrast-enhanced computed tomography (CT) should be obtained at 1 and 12 months after TEVAR, then annually thereafter. As a complication, the term “endoleak” refers to persistent flow into the aneurysm sac, which can happen as a result of the proximal or distal endovascular graft components failing to seal flow at the aneurysm neck, retrograde flow through aortic branches (such as the intercostal, lumbar, or visceral arteries), flow between various graft components, or flow through graft fabric defects. Endoleak following TEVAR has an incidence of 4–15%. It can be detected after surgery (primary) or during surveillance (secondary). Our patient underwent a CT scan one month after TEVAR, which revealed a type II endoleak. A CT scan follow-up should be performed yearly, and if aneurysm sac enlargement is identified, reintervention may be indicated [18,19].

### 3.3. Prognosis

Although conservative management may prevent aorta-related deaths in specific cohorts, the mortality rate for PAU varies but can be substantial. Some studies report an acute PAU in-hospital mortality of 26.7%, and the long-term mortality for isolated PAU is approximately 30% [20]. High-risk characteristics are linked to an increased likelihood of negative outcomes, such as aortic-related death. These characteristics include the presence of a combined intramural hematoma (IMH) and PAU, a deep ulcer (e.g., >9.5 mm), or a large PAU diameter (e.g., >12.5 mm) [21]. While age, comorbidities, and the presence of IMH are essential predictors of death, endovascular repair is linked to a lower in-hospital mortality rate than medical therapy for ascending PAUs. In a trial of 18 patients, there were two reinterventions within 30 days due to type I or III endoleak. Seven of the 18 patients died during follow-up (mean survival, 90.24 months; range, 66.48–113.88), with one having a verified aortic-related mortality [15]. To avoid complications, all patients should have ongoing medical therapy for blood pressure control, as well as regular follow-up and imaging.

## 4. Conclusions

Early detection of a penetrating aortic ulcer is crucial for preventing life-threatening complications. Regular annual monitoring enables timely intervention, improving patient outcomes. Clinicians should remain vigilant for conditions that mimic aortitis, such as infectious or autoimmune diseases, to ensure accurate diagnosis and management. With prompt identification and appropriate care, most individuals with a penetrating aortic ulcer or intramural hematoma can maintain stable health and avoid serious complications.

## Figures and Tables

**Figure 1 diagnostics-15-03077-f001:**
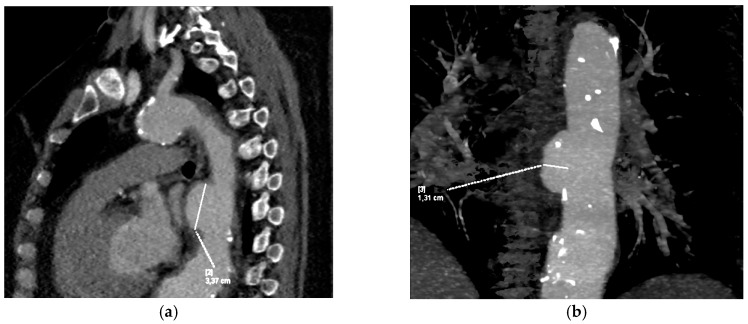
CT angiography showing (**a**) penetrating aortic ulcer (PAU) and (**b**) dilated thoracic aorta.

**Figure 2 diagnostics-15-03077-f002:**
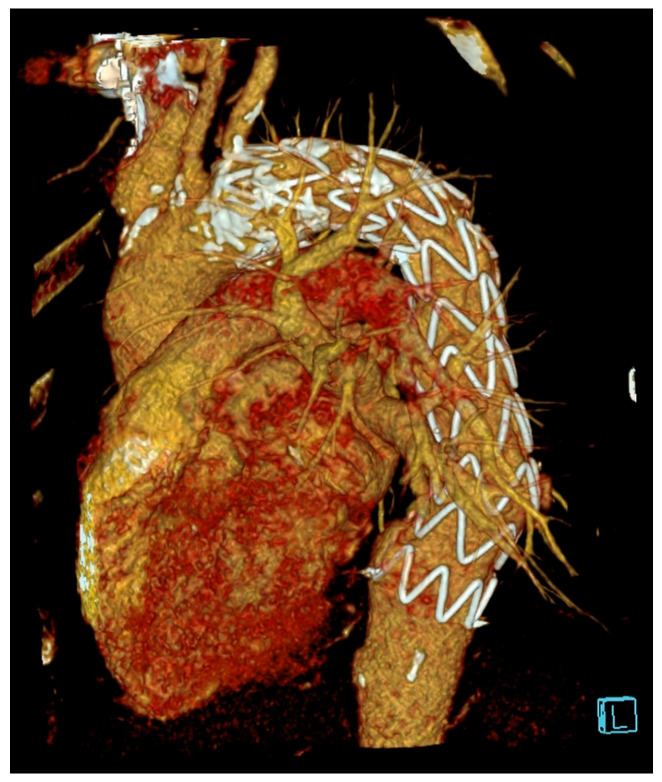
Postprocedural 3D-CTA imaging of the thoracic aorta following TEVAR (Thoracic Endovascular Aortic Repair). The letter L is Left side of the patient.

## Data Availability

The data presented in this study are available on request from the corresponding author due to availability of images.

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
