# Peer review of "Fever of Unknown Origin and Penetrating Aortic Ulcer Successfully Treated with Thoracic Endovascular Aortic Repair—A Case Report"

_diagnostics, 2025, doi:10.3390/diagnostics15233077_

Round 1

Reviewer 1 Report

Comments and Suggestions for Authors

This is an interesting case report of a penetrating aortic ulcer presenting as a pyrexia of unknown origin. There is a logical approach to the appropriate investigations to request and the way other potential conditions are ruled out is exemplary. The choice of imaging is important and this has addressed well in this report. The treatment is clearly described including the management of the patient following the procedure. A lot of abbreviations are used and the list provided is invaluable. 

Author Response

Reviewer 1: This is an interesting case report of a penetrating aortic ulcer presenting as a pyrexia of unknown origin. There is a logical approach to the appropriate investigations to request and the way other potential conditions are ruled out is exemplary. The choice of imaging is important and this has addressed well in this report. The treatment is clearly described including the management of the patient following the procedure. A lot of abbreviations are used and the list provided is invaluable.

We thank the reviewer for his observations.

Reviewer 2 Report

Comments and Suggestions for Authors

While the case is interesting I would like to make the following suggestions.

  1. The title should better highlight the findings, perhaps give an insight into the final diagnosis.
  2. In the abstract, the Introduction is too short and direct.
  3. Please include all reference values for all parameters if you insist on including them, or for none.
  4. Also, either describe the laboratory parameters as incresed or decreased or the change they translate (anemia, thrombocytosis) in the same way for all, but not in one way for some and another for the others.
  5. After how many days was the decrease in CRP noticed?
  6. Also the conclusion is too frank in the abstract.
  7. While the abstract introduction focused on fever as a non-specific sign in patients, the intruduction of the main text focuses directly on aneurisms. Please include in both introduction both informations and perhaps expand on the proper protocol for patients with fever of unknown origin.
  8. Also, the paragraphs in the introduction are not well connected with one-another, the information should flow better.
  9. In the history section, why were the laboratory values included? Please place them seperately and perhaps in a more schematic manner (table, figure).
  10. Also the smoking should be quatified with the pack-year formula (number of years of smoking x the number of cigarettes per day/20).
  11. if you mentioned that perindopril is an ace inhibitor and amlodipine is a calcium channel blocker, you should also mention that warfarin is an anticoagulant for a more homogeneous text.
  12. The term back pain is a bit vague, perhaps describe the exact location and possible radiation, along with the circumstances of the pain (at rest, on effort).
  13. In the discussions I would like to see the AHA and ESC guideline recommendations on aortic disease management.
  14. The conclusions are again too short.
  15. The manuscript is sometimes difficult to read and some paragraphs are not well connected between them. Also, not all abbreviations are explained first time. Please revise the formulations.
Comments on the Quality of English Language

The manuscript is sometimes difficult to read and some paragraphs are not well connected between them. Also, not all abbreviations are explained first time.

Author Response

Reviewer: The title should better highlight the findings, perhaps give an insight into the final diagnosis.

We have added a more informative title.

Reviewer: In the abstract, the Introduction is too short and direct.

We thank the reviewer for this observation. Unfortunately, the abstract is limited to 200 words, which is why it must be concise.

Reviewer: Please include all reference values for all parameters if you insist on including them, or for none.

According to CARE case report guidelines, we added all values through the text. Immune antibodies ANA and ANCA are negative; if they were positive, then the results would be presented with a value and a subtype of antibody.

Reviewer:  Also, either describe the laboratory parameters as increased or decreased or the change they translate (anemia, thrombocytosis) in the same way for all, but not in one way for some and another for the others. After how many days was the decrease in CRP noticed?

We have reported laboratory values according to CARE case report guidelines.

Reviewer:  Also the conclusion is too frank in the abstract.

We thank the reviewer for this observation. Unfortunately, the abstract is limited to 200 words, which is why it must be concise.

Reviewer: While the abstract introduction focused on fever as a non-specific sign in patients, the intruduction of the main text focuses directly on aneurisms. Please include in both introduction both informations and perhaps expand on the proper protocol for patients with fever of unknown origin.

In the introduction, we clearly  stated that fever and general symptoms are part of large vessel vasculitis: “ Aortitis in the setting of vasculitis is usually accompanied by general symptoms, fever, … “We focus on aneurysms because they are the outcome of both vasculitis and atherosclerosis, and the basis of this case report.

Reviewer:  Also, the paragraphs in the introduction are not well connected with one-another, the information should flow better.

We consulted native English speaker and rearranged this paragraph accordingly.

Reviewer: In the history section, why were the laboratory values included? Please place them seperately and perhaps in a more schematic manner (table, figure).

The case report is written according to CARE check list.

Reviewer: Also the smoking should be quatified with the pack-year formula (number of years of smoking x the number of cigarettes per day/20).

We added a formula that is suggested by a pulmonologist (Number of packs smoked per day) x (Number of years smoked).

40 pack – years.

Reviewer:  if you mentioned that perindopril is an ace inhibitor and amlodipine is a calcium channel blocker, you should also mention that warfarin is an anticoagulant for a more homogeneous text.

Warfarin is mentioned in the same sentence with other drugs.

Reviewer: The term back pain is a bit vague, perhaps describe the exact location and possible radiation, along with the circumstances of the pain (at rest, on effort).

We added more information about her back pain.

Reviewer:  In the discussions I would like to see the AHA and ESC guideline recommendations on aortic disease management.

Isselbacher, E.M.; Preventza, O.; Hamilton Black, J., 3rd; Augoustides, J.G.; Beck, A.W.; Bolen, M.A.; Braverman, A.C.; Bray, B.E.; Brown-Zimmerman, M.M.; Chen, E.P.; et al. 2022 ACC/AHA Guideline for the Diagnosis and Management of Aortic Disease: A Report of the American Heart Association/American College of Cardiology Joint Committee on Clinical Practice Guidelines. Circulation 2022, 146, e334–e482.

Reviewer:  The conclusions are again too short.

We rearranged the conclusions to be more informative  with a clear message to reader.

Reviewer:  The manuscript is sometimes difficult to read and some paragraphs are not well connected between them. Also, not all abbreviations are explained first time. Please revise the formulations.

The journal requirements propose subtitles and shortened paragraphs. We tried not to clutter the text.

Round 2

Reviewer 2 Report

Comments and Suggestions for Authors

While several comments have been addressed, I still find that some were skipped.

My indications were given in the idea of uniformity.

It is better to present all medication in the same way. Instead you opted to mention either a class of medication (Ace inhibotors) while the others as active substance. And I quote the sentence: 

”Her previous treatment included warfarin 64 and a combination of an ACE inhibitor (perindopril 4 mg) with a calcium channel blocker 65 (amlodipine 5 mg). ”

In this sentence in particular you mention the class of medication for perindopril and for amlodipine and it would be more uniform to mention it for warfarin.

Secondly I understand that there are limits to the number of words in the abstract, yet it can be rephrased to better highlight the findings of the study.

Please revise my previous comments and address them accordingly.

Comments on the Quality of English Language

The manuscript is sometimes difficult to read and some paragraphs are not well connected between them. Also, not all abbreviations are explained first time.

Author Response

REVIEWER: several comments have been addressed, I still find that some were skipped.

Thank you for your observation. We have highlighted the new revision in green. New revision included previous observations regarding abstract, CRP, clarity and text flow.

REVIEWER: My indications were given in the idea of uniformity. It is better to present all medication in the same way. Instead you opted to mention either a class of medication (Ace inhibotors) while the others as active substance. And I quote the sentence: ”Her previous treatment included warfarin 64 and a combination of an ACE inhibitor (perindopril 4 mg) with a calcium channel blocker 65 (amlodipine 5 mg). ” In this sentence in particular you mention the class of medication for perindopril and for amlodipine and it would be more uniform to mention it for warfarin.

We have added anticoagulants (warfarin) to the bin in line with the reporting of other drugs.

REVIEWER: Secondly I understand that there are limits to the number of words in the abstract, yet it can be rephrased to better highlight the findings of the study.

We have written an entirely new summary.

REVIEWER: Please revise my previous comments and address them accordingly.

We have highlighted the new revision in green. New revision included previous observations regarding abstract, CRP, clarity, conclusion, and text flow.